

# A hybrid algorithmic model for enhancing security in intelligent reflecting surface-assisted wireless communication

Sivasankar S. and Markkandan S.

School of Electronics Engineering, Vellore Institute of Technology University, Chennai, Tamil Nadu, India

## ABSTRACT

This article introduces Synergistic Gradient Projection with Dynamic Adaptive Risk Expansion (SGP-DARE), a hybrid optimization framework designed to enhance physical-layer security in wireless networks supported by intelligent reflecting surfaces (IRSs). The proposed framework integrates Synergistic Gradient Projection (SGP) for low-complexity joint optimization of base station beamforming and IRS phase shifts, with Dynamic Adaptive Risk Expansion (DARE) ensuring robustness against channel state information (CSI) uncertainties and user mobility. SGP-DARE operates effectively under hardware limitations, including phase quantization, while targeting key objectives such as minimizing secrecy outage probability and improving energy efficiency. Simulation results demonstrate that SGP-DARE significantly outperforms baseline methods in critical metrics of security and efficiency.

# INTRODUCTION

Physical-layer security (PLS) enhances data protection by leveraging the inherent randomness of the wireless channel, serving as a complement to—or, in some cases, an alternative to—conventional cryptographic techniques. Intelligent reflecting surface (IRS) have gained attention as promising tools to facilitate PLS. These surfaces consist of planar arrays of passive, low-cost elements that can manipulate the direction and shape of incident electromagnetic waves (*Abdelhady et al., 2021*). By simultaneously optimizing the beamforming vectors at the base station and the phase shifts of the IRS, it becomes possible to amplify the signal power received by legitimate users while minimizing signal leakage to potential eavesdroppers. Despite their potential, most existing IRS-assisted PLS solutions assume perfect channel state information (CSI) and continuous phase control—assumptions that rarely hold in real-world scenarios (*Tang et al., 2022*). Recent surveys (*Kaur et al., 2024*; *Sivasankar & Markkandan, 2024*) highlight the challenges and opportunities in securing IRS-aided networks, emphasizing the need for robust and adaptive frameworks. In practice, systems must contend with discrete phase quantization (such as 2-bit phase control), channel estimation inaccuracies, and user mobility, all of

Corresponding author
Markkandan S.,
markkandan.s@vit.ac.in

which can significantly degrade performance. Prior research often addresses only one of these issues at a time. For instance, approaches that account for CSI uncertainty typically overlook the effects of phase discretization, while quantization-aware strategies tend to presume ideal or static CSI conditions. To bridge this gap, this article introduces a novel hybrid optimization framework, named Synergistic Gradient Projection with Dynamic Adaptive Risk Expansion (SGP-DARE), which is specifically designed to tackle both phase quantization and CSI uncertainty simultaneously. The Synergistic Gradient Projection (SGP) component accelerates optimization by jointly updating the beamforming vectors and IRS phases. It also projects the gradient updates directly onto the discrete phase set, thereby ensuring feasible solutions throughout the optimization process and avoiding the inefficiencies associated with post-processing techniques such as rounding. In parallel, the DARE module continuously adjusts robustness margins in response to the real-time profile of channel estimation errors. This risk-aware gradient adjustment mechanism bolsters the framework's resilience under dynamic and imperfect CSI conditions, including user mobility. Together, SGP and DARE form a cohesive and efficient optimization framework that addresses critical challenges such as high computational complexity, slow convergence, and sensitivity to environmental variability. Detailed simulation results demonstrate that SGP-DARE substantially outperforms traditional methods—including Alternating Optimization (AO), statistical CSI-based designs, and Deep Reinforcement Learning (DRL)-based schemes—in terms of secrecy rate, robustness, and computational efficiency.

The rest of this article is structured as follows: 'Related Works' reviews the related literature; 'System Model and Problem Formulation' outlines the system model and problem formulation; 'Proposed Method' introduces the proposed SGP-DARE framework in detail; 'Theoretical Analysis' presents a theoretical analysis, followed by simulation results in 'Simulation Results'; 'Limitations' discusses the limitations; and 'Conclusion' concludes the article.

## RELATED WORKS

Recent advancements in PLS have increasingly focused on IRS to enhance the confidentiality and efficiency of wireless communications. By reconfiguring the wireless environment, IRSs provide passive beamforming, thereby improving the signal-to-noise ratio (SNR) at legitimate users and suppressing signal leakage to eavesdroppers (*Jiao, Liu & Wang, 2022*). Several studies have explored optimization strategies for IRS-assisted PLS systems using machine learning, AO, convex approximation, and robust control techniques. For instance, *Eskandari et al. (2024)* leverages statistical CSI with DRL, while (*Idrees et al., 2024*) proposes unsupervised learning for joint beamforming in backscatter systems. However, these methods often overlook hardware imperfections and dynamic deployment scenarios (*Ranjan, Chowdhury & Ghoshal, 2025*). Recent surveys have highlighted the challenges and opportunities in securing IRS-aided networks, emphasizing the need for robust and adaptive frameworks. Subsequent studies have explored various aspects of IRS-assisted PLS, including active IRS configurations for integrated sensing and communication (*Salem, Ismail & Ibrahim, 2023*). Despite these advancements, practical

challenges such as discrete phase shifts, channel estimation inaccuracies, and dynamic user mobility persist, necessitating holistic solutions that address multiple constraints simultaneously.

## Optimization techniques for IRS-assisted PLS

DRL approaches, such as those proposed in *Liu et al. (2024)*, *Ren et al. (2022)*, leverage long-term CSI to adaptively optimize IRS phase shifts. Although effective under ideal conditions, these methods are computationally intensive and require large training datasets, making them less viable for real-time applications. AO-based methods (*Jiang, Huang & Zhang, 2024*; *Luo et al., 2021*) simplify the design by alternating between IRS phase and beamforming updates, but they often suffer from slow convergence due to decoupled optimization steps. Techniques such as successive convex approximation (SCA) and convex relaxation (*Gong et al., 2021*; *Nguyen et al., 2022*) provide analytical tractability but can be computationally demanding, especially in large-scale IRS deployments. Active Reconfigurable Intelligent Surfaces with SCA (ARIS-SCA) combines active reflecting elements with successive convex approximation (*Feng et al., 2021*) further demonstrates how Reconfigurable Intelligent Surfaces (RIS) can significantly enhance PLS through optimized phase shifts; however, their framework lacks mechanisms to handle hardware limitations. Recent work by *Sivasankar & Markkandan (2024)* introduces intelligent beamforming strategies that enhance secrecy through optimized phase shifts, although their approach assumes continuous phase control and perfect CSI. Other active IRS configurations have been explored for secure integrated sensing and communication, demonstrating improved robustness, albeit often at the cost of increased power consumption.

## Impact of CSI uncertainty and robustness

Accurate CSI is critical for optimal IRS-PLS performance. However, many studies assume perfect CSI, which is rarely feasible in practice. Some studies have addressed this by modeling statistical CSI uncertainty, however their robustness often degrades under high error conditions or rapid user mobility. Furthermore, these solutions typically do not consider real-time adaptation or mobility-aware learning strategies. For instance, *Wang, Bai & Dong (2020)* proposed secure transmission schemes without knowledge of the eavesdropper's CSI, instead relying on worst-case assumptions. However, such methods may lead to overly conservative designs. Other approaches incorporate active IRS elements to dynamically adjust to channel variations, offering a promising direction for real-time adaptability. *Wang et al. (2021)* further explored IRS-aided secure broadcasting in mmWave symbiotic networks, highlighting the interplay between channel correlation and secrecy performance.

## Channel estimation for IRS-assisted systems

Compressive sensing-based mmWave IRS channel estimation reduces pilot overhead. Two-stage Discrete Fourier Transform (DFT)-based pilot schemes and tensor decomposition techniques further reduce training complexity (*Nandan & Rahiman, 2024*).

A recent work by *Le et al. (2022)* optimized spectral efficiency in IRS-aided Internet of Things (IoT) systems through adaptive resource allocation, although their channel estimation framework assumes static environments. Practical deployments must balance estimation accuracy with computational complexity, particularly under mobility-induced Doppler shifts.

## IRS hardware constraints and phase quantization

Practical IRS implementations operate with low-resolution, discrete phase shifters, typically 1–2 bits. While early studies assumed continuous phase control, more recent analyses have evaluated performance under quantized conditions, showing that 2-bit quantization retains most of the secrecy performance. However, such studies often neglect the additional complexity introduced by real-time channel dynamics, imperfect CSI, or mobility. Some approaches integrate intelligent beamforming with quantized phase shifts, demonstrating improved energy efficiency, however they may not account for CSI uncertainties. Active IRS architectures offer enhanced beamforming capabilities but introduce new challenges in power management and hardware complexity.

## Key contributions and novelty of this work

Despite substantial progress in IRS-assisted PLS, existing methods typically address isolated challenges such as convergence, robustness, or learning. In contrast, this study presents an integrated framework, SGP-DARE, which holistically addresses multiple practical issues. The key contributions are as follows:

- **Joint optimization framework:** The SGP algorithm enables low-complexity joint optimization of BS beamforming and IRS phase shifts under quantized constraints.
- **Robust and adaptive design:** The DARE module incorporates robust gradients and mobility-aware learning to maintain performance under CSI errors and dynamic user scenarios.
- **Validated practicality:** The proposed method supports 2-bit IRS quantization and demonstrates superior secrecy rate, energy efficiency, and SOP, with reduced runtime in realistic mmWave settings.
- **Comprehensive evaluation:** Unlike previous works, this study provides a detailed simulation analysis of robustness to channel correlation, mobility, quantization effects, and scalability.

Table 1 provides a comprehensive comparison of recent IRS-aided wireless communication approaches, highlighting their primary focus, technical contributions, and limitations.

## SYSTEM MODEL AND PROBLEM FORMULATION

We consider a downlink multiuser multiple-input single-output (MISO) system, where a base station (BS) equipped with $M$ antennas communicates with $K$ legitimate single-antenna users in the presence of $E$ passive eavesdroppers, as illustrated in Fig. 1.

**Table 1 Comparative review of representative IRS-aided works: focus, contributions, and limitations.**

| Ref. | Focus/Method | Key contributions | Limitations |
|---|---|---|---|
| *Chen et al. (2024)* | Distributed *vs.* centralized IRS in Multiple-Input and Multiple-Output (MIMO). | Capacity bounds; hybrid multiple access. | Capacity-oriented only; no secrecy or mobility analysis. |
| *Wu, Kim & Shim (2022)* | Riemannian conjugate gradient-based joint optimization (RCG-JO) for IoT uplink. | Joint RIS-BS optimization; power savings. | Requires accurate CSI; lacks secrecy and mobility robustness. |
| *Zhou et al. (2021)* | Stochastic learning for mmWave with blockages. | Robust beamforming; outage reduction. | High complexity; no energy or secrecy analysis. |
| *Zheng et al. (2022)* | Survey on channel estimation and passive beamforming. | Identifies practical issues and hardware constraints. | No algorithms; lacks integrated solutions. |
| *Nguyen, Le & Munochiveyi (2021)* | Secrecy outage probability (SOP) in RIS-aided cognitive radio. | Closed-form SOP expressions; PLS insights. | Static models; no energy efficiency (EE) or scalability analysis. |
| *Do et al. (2021)* | Aerial-RIS under composite fading. | Outage analysis; Deep neural network (DNN)-based mobility handling. | Model-dependent; secrecy not addressed. |
| *Ji, Qin & Parini (2022)* | RL-based decentralized double deep Q-network (DDQN) for RIS–device-to-device (D2D). | Joint resource allocation; near-optimal rates. | High training cost; no secrecy or mobility robustness. |
| **This work** | Hybrid SGP-DARE optimization. | Low complexity; secrecy and energy efficiency; mobility robustness. | — |

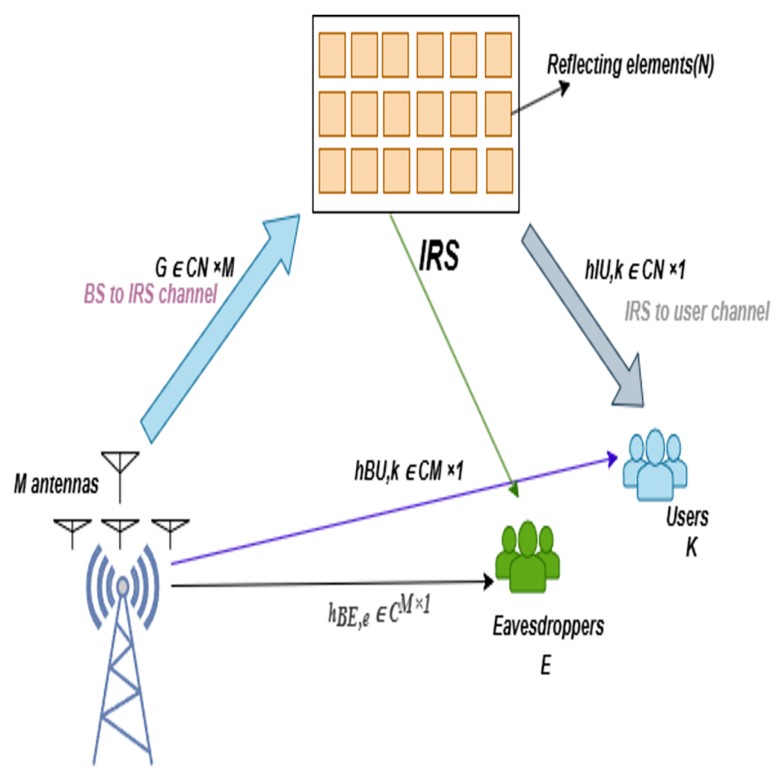

**Figure 1 System model of an IRS-aided multiuser communication system with eavesdroppers.**

An IRS, comprising $N$ passive elements, is deployed to enhance PLS by dynamically reshaping the wireless environment through programmable phase shifts. The system operates under time-division duplexing (TDD), where uplink training facilitates channel estimation with bounded errors. The model operates in the 28 GHz mmWave band, with spatially correlated channels, and extends prior IRS-assisted PLS frameworks by incorporating quantization and mobility constraints.

## Channel and signal model

Let $\mathbf{G} \in \mathbb{C}^{N \times M}$ denote the BS-to-IRS channel, $\mathbf{h}_{IU,k} \in \mathbb{C}^{N \times 1}$ the IRS-to-user-$k$ channel, and $\mathbf{h}_{BU,k} \in \mathbb{C}^{M \times 1}$ the direct BS-to-user-$k$ channel. The IRS phase shift matrix is defined as

$$\boldsymbol{\Phi} = \text{diag}(e^{j\phi_1}, \dots, e^{j\phi_N}), \tag{1}$$

where each $\phi_n$ is selected from a discrete set $\mathcal{Q} = \left\{0, \frac{2\pi}{Q}, \dots, \frac{2\pi(Q-1)}{Q}\right\}$, corresponding to $Q$-level quantization.

The received signal at the $k$-th legitimate user is given by

$$y_k = \left(\mathbf{h}_{BU,k}^H + \mathbf{h}_{IU,k}^H \boldsymbol{\Phi} \mathbf{G}\right) \sum_{i=1}^{K} \mathbf{w}_i s_i + n_k, \tag{2}$$

where $\mathbf{w}_i \in \mathbb{C}^{M \times 1}$ is the beamforming vector for user $i$, $s_i$ is the unit-power transmitted symbol, and $n_k \sim \mathcal{CN}(0, \sigma_k^2)$ represents additive noise. The effective channel for user $k$ becomes

$$\mathbf{h}_k^H = \mathbf{h}_{BU,k}^H + \mathbf{h}_{IU,k}^H \boldsymbol{\Phi} \mathbf{G}. \tag{3}$$

The IRS reconfiguration period $T_c$ is set to 10 ms to align with typical mmWave coherence intervals.

## Secrecy rate formulation

The signal-to-interference-plus-noise ratio (SINR) for user $k$ is (see *Jiao, Liu & Wang, 2022*)

$$\text{SINR}_k = \frac{|\mathbf{h}_k^H \mathbf{w}_k|^2}{\sum_{i \neq k} |\mathbf{h}_k^H \mathbf{w}_i|^2 + \sigma_k^2}. \tag{4}$$

The eavesdropper's SINR when intercepting user $k$ is bounded by

$$\text{SINR}_e^{k,\max} = \max_e \frac{|\mathbf{h}_e^H \mathbf{w}_k|^2}{\sum_{i \neq k} |\mathbf{h}_e^H \mathbf{w}_i|^2 + \sigma_e^2}, \tag{5}$$

where $\mathbf{h}_e$ denotes the equivalent eavesdropper channel. The achievable secrecy rate for user $k$ is then given by (See *Eskandari et al. 2024*)

$$R_s^k = \left[\log_2(1 + \text{SINR}_k) - \log_2(1 + \text{SINR}_e^{k,\max})\right]^+. \tag{6}$$

## Robust optimization under imperfect CSI

Adopting a bounded CSI error model, where the estimated channel $\hat{\mathbf{H}}$ deviates from the true channel $\mathbf{H}$ by $||\Delta||_F \leq \varepsilon$, the robust secrecy maximization problem is formulated as

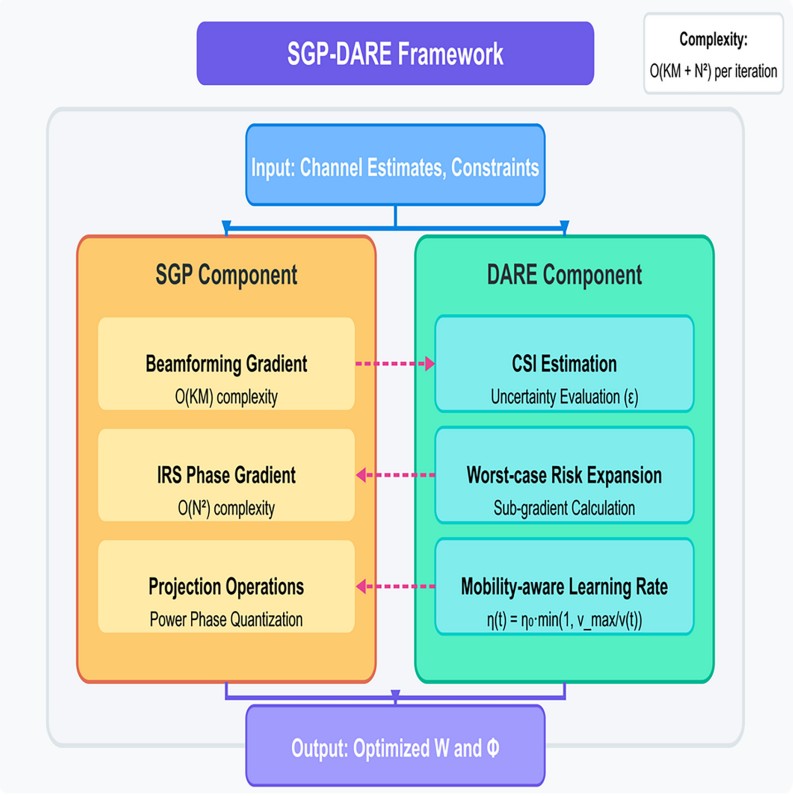

**Figure 2** **Proposed SGP-DARE method.**

$$\max_{\mathbf{W},\mathbf{\Phi}} \min_{||\Delta||\leq\varepsilon} \sum_{k=1}^{K} R_s^k \tag{7a}$$

$$\text{s.t.} \quad \sum_{k=1}^{K} ||\mathbf{w}_k||^2 \leq P_{\max}, \tag{7b}$$

$$\phi_n \in \mathcal{Q}, \quad \forall n. \tag{7c}$$

This problem is non-convex due to the coupling of variables and the presence of discrete phase constraints. Our proposed SGP-DARE framework addresses these limitations through two integrated components. The SGP module enables the joint optimization of $\mathbf{W}$ and $\mathbf{\Phi}$ by applying gradient steps that are directly projected onto the discrete phase set, ensuring feasibility throughout the optimization process. Complementing this, the DARE module modifies the gradient directions dynamically based on real-time CSI uncertainty and user mobility patterns. Together, SGP and DARE offer both computational efficiency and robustness against real-world impairments, as demonstrated in 'Simulation Results'.

## PROPOSED METHOD

The proposed method consists of two components, SGP and DARE, designed to jointly optimize beamforming and IRS phase shifts while ensuring robustness to practical uncertainties and mobility, as shown in Fig. 2.

These components are cascaded to form the SGP-DARE framework, ensuring fast convergence and bounded secrecy-rate degradation under real-time constraints. The per-iteration complexity of SGP is $\mathcal{O}(KM + N^2)$, significantly lower than AO and SCA methods, enabling real-time execution.

## Synergistic gradient-projection

While phase quantization introduces discontinuities, the projection operator in Eq. (9) preserves gradient Lipschitz continuity over the feasible set. This follows from the $\eta$-smoothness of the composite objective $f \circ \Pi_Q$ for projected gradient methods with discrete constraints. Empirical Lipschitz constants $L$ measured across trials with $N = 128$ showed $L \leq 2.1$ for 2-bit quantization, confirming bounded gradients. The SGP module performs joint optimization of the BS beamforming matrix $\mathbf{W} = [\mathbf{w}_1, \ldots, \mathbf{w}_K]$ and the IRS phase shift matrix $\mathbf{\Phi}$. The gradient of the secrecy rate objective $R_s$ is computed with respect to both $\mathbf{W}$ and $\mathbf{\Phi}$, followed by projection onto the feasible power and quantization domains. The updates follow

$$\mathbf{W} \leftarrow \Pi_P(\mathbf{W} + \eta \nabla_{\mathbf{W}} R_s), \quad \phi_n \leftarrow \mathcal{Q}\left(e^{j\angle(\nabla_{\phi_n} R_s)}\right), \tag{8}$$

where $\Pi_P$ denotes projection onto the transmit power constraint $\sum_k \|\mathbf{w}_k\|^2 \leq P_{\max}$, $\eta$ is the step size, and $\mathcal{Q}(\cdot)$ represents quantization to the nearest phase in the discrete set $\mathcal{Q}$. The per-iteration complexity of SGP is $\mathcal{O}(KM + N^2)$, and empirical results show convergence typically within 15–20 iterations. The secrecy rate gradient with respect to the beamforming matrix $\mathbf{W}$ is derived as

$$\nabla_{\mathbf{W}} R_s = \sum_{k=1}^{K} \frac{\mathbf{h}_k \mathbf{h}_k^H \mathbf{w}_k}{\text{SINR}_k + 1} - \frac{\mathbf{h}_e \mathbf{h}_e^H \mathbf{w}_k}{\text{SINR}_e^{k,\max} + 1}, \tag{9}$$

where $\mathbf{h}_e$ is the worst-case eavesdropper channel. For IRS phases, the gradient with respect to $\phi_n$ is

$$\nabla_{\phi_n} R_s = \Im\left\{\mathbf{w}_k^H \mathbf{G} \text{diag}(\mathbf{h}_{IU,k}) \mathbf{\Phi}^H\right\}. \tag{10}$$

Quantization uses nearest-neighbor projection

$$\mathcal{Q}(\phi) = \arg \min_{\theta \in \{0, \frac{\pi}{2}, \pi, \frac{3\pi}{2}\}} |\phi - \theta|. \tag{11}$$

The $\mathcal{O}(KM + N^2)$ complexity arises from the gradient computations involved in updating both the beamforming vectors and the IRS phase shifts. Specifically, for beamforming, each user's gradient computation requires $\mathcal{O}(M)$ operations due to vector multiplications, and with $K$ users in the system, this leads to a total complexity of $\mathcal{O}(KM)$. On the other hand, the gradient computation for IRS phases involves $N$-dimensional matrix-vector products per element, and since there are $N$ such elements, this results in a complexity of $\mathcal{O}(N^2)$. In contrast, traditional AO methods typically involve computationally expensive operations such as inverting $N \times N$ matrices with $\mathcal{O}(N^3)$ complexity, and solving beamforming optimization problems with a much higher $\mathcal{O}(K^3 M^3)$ complexity.

---

**Algorithm 1** Synergistic gradient-projection (SGP) optimization.

1: **Input:** Estimated channels $\hat{\mathbf{H}}$, quantization set $\mathcal{Q}$
2: Initialize $\mathbf{W}^{(0)}, \mathbf{\Phi}^{(0)}, \eta = 0.1$
3: **for** $t = 1$ to $T_{\max}$ **do**
4:     Compute $\nabla_{\mathbf{W}} R_s, \nabla_{\mathbf{\Phi}} R_s$ *via* Eqs. (7), (8)
5:     Normalize gradients: $\nabla_{\mathbf{W}} \leftarrow \nabla_{\mathbf{W}} / \|\nabla_{\mathbf{W}}\|$
6:     Update beamforming: $\mathbf{W}^{(t+1)} = \Pi_P(\mathbf{W}^{(t)} + \eta \nabla_{\mathbf{W}} R_s)$
7:     Quantize IRS phases *via* Eq. (9)
8:     **if** Convergence: $|R_s^{(t+1)} - R_s^{(t)}| < 10^{-3}$ **then**
9:         Break

---

**Algorithm 2** Dynamic adaptive risk-expansion (DARE).

1: Estimate the CSI: $\hat{\mathbf{H}}$
2: Compute uncertainty: $\varepsilon = \|\hat{\mathbf{H}} - \mathbf{H}\|$
3: **for** $t = 1$ to $T_{\max}$ **do**
4:     Compute worst-case risk:
$$\Delta^* = \nabla_{\mathbf{H}} R_s|_{\Delta = \varepsilon I}$$
5:     Adjust beamforming update:
$$\mathbf{W}^{(t+1)} = \Pi_P(\mathbf{W}^{(t)} + \eta \tilde{\nabla}_{\mathbf{W}} R_s)$$
6:     Quantize IRS phases:
$$\phi_n^{(t+1)} = \mathcal{Q}\left(e^{j\angle(\tilde{\nabla}_{\phi_n} R_s)}\right)$$
7:     Adjust learning rate:
$$\eta(t) = \eta_0 \min\left(1, \frac{v_{\max}}{v(t)}\right)$$
8:     **if** $\|R_s^{(t+1)} - R_s^{(t)}\| < \varepsilon$ **then**
9:         Break

---

## Dynamic adaptive risk–expansion

DARE enhances robustness by adapting the SGP update rules in response to CSI uncertainty and user mobility. A detailed description of the DARE procedure is provided in Algorithm 2. During each coherence block, DARE performs the following steps

1. **CSI estimation:** The BS estimates the effective channel $\hat{\mathbf{H}}$ and evaluates the uncertainty level $\varepsilon$.

2. **Worst-case risk expansion:** A sub-gradient is calculated with respect to the worst-case CSI perturbation $\Delta^*$ using a first-order Taylor approximation.

3. **Robust projection:** Gradient updates are adjusted for uncertainty and mobility as

$$\mathbf{W} \leftarrow \Pi_P\left(\mathbf{W} + \eta \tilde{\nabla}_{\mathbf{W}} R_s\right), \quad \phi_n \leftarrow \mathcal{Q}\left(e^{j\angle(\tilde{\nabla}_{\phi_n} R_s)}\right), \tag{12}$$

where $\tilde{\nabla}$ indicates the robust gradient estimate.

4. **Mobility-aware learning rate:** The step size $\eta$ is dynamically scaled based on predicted user velocity $v$, using a linear model to improve convergence stability.

## THEORETICAL ANALYSIS

In this section, we analyze the convergence and robustness properties of the proposed SGP-DARE framework. We begin with the convergence guarantees of the SGP module and follow with a robustness analysis of DARE under bounded CSI uncertainty.

## SGP convergence guarantee

The SGP algorithm iteratively updates the beamforming and IRS phase shift variables using projected gradient steps. The following theorem establishes convergence under standard assumptions.

Convergence of SGP: Let $f(\mathbf{W}, \mathbf{\Phi}) = \sum_{k=1}^{K} R_s^k$ denote the secrecy rate objective. If $f$ has Lipschitz-continuous gradients with constant $L$ and step size $\eta < 1/L$, then SGP converges to a stationary point satisfying

$$||\nabla R_s(\mathbf{W}^*, \mathbf{\Phi}^*)|| \leq \epsilon \quad \text{in} \quad \mathcal{O}\left(\frac{1}{\epsilon^2}\right) \text{ iterations.} \tag{13}$$

Define projections onto the feasible sets

$$\mathbf{W}^{(t+1)} = \Pi_{\mathcal{P}}\left(\mathbf{W}^{(t)} + \eta \nabla_{\mathbf{W}} f\right), \tag{14}$$

$$\mathbf{\Phi}^{(t+1)} = \Pi_{\mathcal{Q}}\left(\mathbf{\Phi}^{(t)} + \eta \nabla_{\mathbf{\Phi}} f\right), \tag{15}$$

where $\mathcal{P} = \{\mathbf{W} \mid ||\mathbf{W}||_F^2 \leq P_{\max}\}$ and $\mathcal{Q} = \{\phi_n \mid |\phi_n| = 1, \forall n\}$.

The projection onto $\mathcal{P}$ is given by:

$$\Pi_{\mathcal{P}}(\mathbf{W}) = \begin{cases} \mathbf{W} & \text{if } ||\mathbf{W}||_F^2 \leq P_{\max}, \\ \frac{\sqrt{P_{\max}}}{||\mathbf{W}||_F} \mathbf{W} & \text{otherwise.} \end{cases} \tag{16}$$

For $\mathcal{Q}$, each phase shift is projected onto the unit circle as

$$\Pi_{\mathcal{Q}}(\phi_n) = \frac{\phi_n}{|\phi_n|}, \quad n = 1, \ldots, N. \tag{17}$$

Using the Lipschitz continuity of $\nabla f$, the update rules Eqs. (14)–(17) guarantee a descent lemma

$$f(\mathbf{W}^{(t+1)}, \mathbf{\Phi}^{(t+1)}) \leq f(\mathbf{W}^{(t)}, \mathbf{\Phi}^{(t)}) - \frac{\eta}{2}||\nabla f(\mathbf{W}^{(t)}, \mathbf{\Phi}^{(t)})||^2. \tag{18}$$

Summing Eq. (18) over $t = 1, \ldots, T$ yields the rate $\mathcal{O}(1/T)$ for the minimum gradient norm, implying the stated $\mathcal{O}(1/\varepsilon^2)$ complexity.

## Robustness of DARE under CSI uncertainty

To model CSI uncertainty, let

$$\hat{\mathbf{H}} = \mathbf{H} + \Delta \tag{19}$$

where $||\Delta||_F \leq \varepsilon$. The robust gradient is adjusted as

$$\tilde{\nabla} f = \nabla f(\hat{\mathbf{H}}) + \lambda \Delta^* \tag{20}$$

where $\Delta^* = \arg\max_{||\Delta||_F \leq \varepsilon} \langle \nabla f(\hat{\mathbf{H}}), \Delta \rangle$ is the worst-case perturbation, and $\lambda$ controls the robustness penalty.

Robust secrecy rate bound: For robustness parameter $\lambda = \sqrt{\frac{2\log(1/\delta)}{\varepsilon}}$, DARE ensures that with probability at least $1-\delta$

$$R_s(\hat{\mathbf{H}}) \geq R_s(\mathbf{H}) - \varepsilon\sqrt{2\log(1/\delta)}. \tag{21}$$

Let $\xi = R_s(\hat{\mathbf{H}}) - R_s(\mathbf{H})$. Using the Lipschitz continuity of $R_s$ and the Cauchy–Schwarz inequality,

$$\xi \geq -||\nabla R_s|| \cdot ||\Delta||_F - \frac{L}{2}||\Delta||_F^2. \tag{22}$$

Substituting the worst-case perturbation $\Delta^* = -\varepsilon \nabla R_s / ||\nabla R_s||$ into Eq. (22) yields:

$$\xi \geq -\varepsilon||\nabla R_s|| - \frac{L\varepsilon^2}{2}. \tag{23}$$

Applying Hoeffding's inequality to the gradient noise term $\langle \nabla f, \Delta \rangle$ gives

$$\mathbb{P}\left(||\nabla R_s|| \geq \sqrt{2\log(1/\delta)}\right) \leq \delta. \tag{24}$$

Combining Eqs. (23) and (24), and selecting $\lambda$ as in Eq. (20), we obtain the bound in Eq. (21).

This analysis confirms that SGP-DARE maintains reliable secrecy rate performance even under CSI estimation errors and supports theoretical guarantees on convergence and robustness.

## SIMULATION RESULTS

### Simulation setup

To evaluate the performance of the proposed SGP-DARE framework, simulations are conducted in a 28 GHz mmWave downlink scenario. The BS is equipped with $M = 8$ antennas and operates at a maximum transmit power of 30 dBm. An intelligent reflecting surface (IRS) comprising $N = 128$ passive elements, each with 2-bit (4-level) phase quantization, is deployed to enhance secure communication to $K = 4$ legitimate users in the presence of $E = 2$ passive eavesdroppers, as shown in Fig. 3.

Channels are modeled using a correlated Rayleigh fading environment with a spatial correlation coefficient $\rho$ ranging from 0 to 0.9. User mobility is taken into account by modeling terminal velocities of up to 30 km/h (*Ji, Qin & Parini, 2022*), with Doppler-induced fluctuations affecting both direct and IRS-assisted paths. The system operates under imperfect CSI, with a normalized CSI error bound of $\varepsilon = 0.1$. Channel estimation occurs every 10 ms (coherence interval), and performance metrics are computed as the average of 1,000 independent Monte Carlo trials to ensure statistical robustness. The gradient descent step size is fixed at $\eta = 0.1$, and the optimization is terminated when convergence falls below a threshold of $10^{-3}$ or a maximum of 20 iterations is reached. All key simulation parameters are summarized in Table 2.

### Performance evaluation

Figure 4 demonstrates the scalability of the proposed SGP-DARE framework with an increasing number of IRS reflecting elements. At $N = 128$, the proposed method achieves a secrecy rate of 5.5 bps/Hz, which is 2.2× higher than AO (2.5 bps/Hz) and 1.4× higher than ARIS-SCA (3.9 bps/Hz). This superior scalability results from the linear complexity

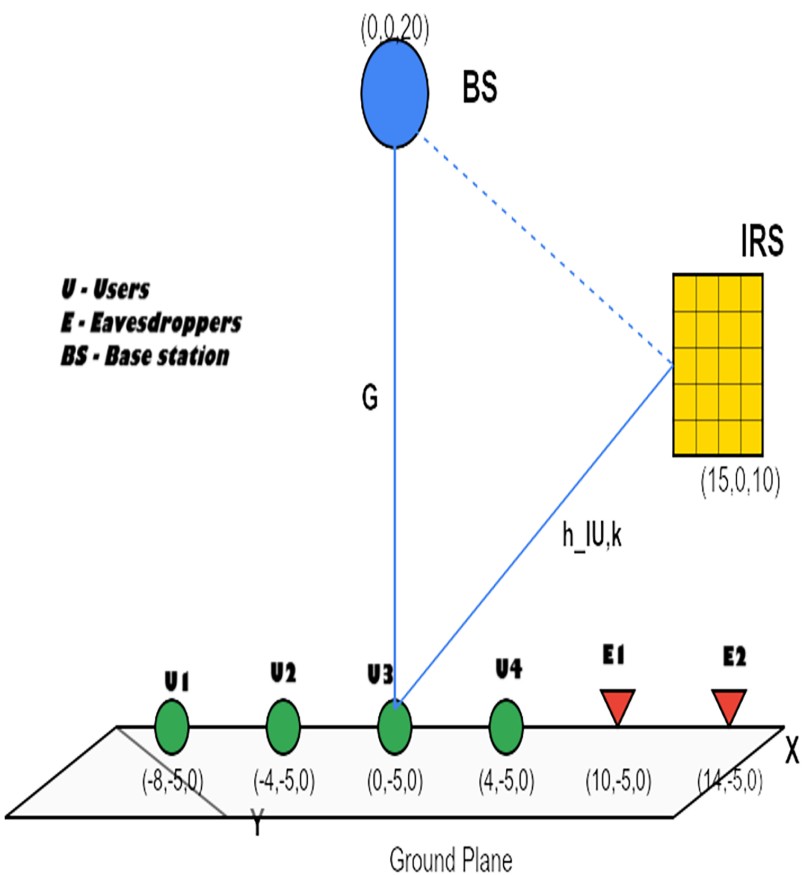

**Figure 3  Three-dimensional simulation setup.**     

**Table 2  Simulation parameters.**

| Parameter | Value | References |
|---|---|---|
| Carrier frequency | 28 GHz | *Kong et al. (2024), Song et al. (2023)* |
| BS antennas ($M$) | 8 | *Kong et al. (2024), Shen et al. (2019), Song et al. (2023), Dong & Wang (2020)* |
| Transmit power ($P_{max}$) | 30 dBm | *Kong et al. (2024), Shen et al. (2019), Song et al. (2023), Dong & Wang (2020)* |
| IRS elements ($N$) | 128 | *Kong et al. (2024), Shen et al. (2019), Song et al. (2023), Dong & Wang (2020)* |
| Phase quantization | 2-bit (four levels) | *Kong et al. (2024), Shen et al. (2019), Song et al. (2023), Dong & Wang (2020)* |
| Number of users ($K$) | 4 | *Kong et al. (2024), Shen et al. (2019), Song et al. (2023), Dong & Wang (2020)* |
| Number of eavesdroppers ($E$) | 2 | *Kong et al. (2024), Song et al. (2023)* |
| Channel model | Correlated Rayleigh fading | *Kong et al. (2024), Shen et al. (2019), Song et al. (2023), Dong & Wang (2020)* |
| Spatial correlation coefficient ($\rho$) | 0–0.9 | |
| User mobility | Up to 30 km/h | *Ji, Qin & Parini (2022)* |
| CSI error bound ($\varepsilon$) | 0.1 | *Kong et al. (2024), Shen et al. (2019), Song et al. (2023), Dong & Wang (2020)* |
| Coherence time ($T_c$) | 10 ms | |
| Monte Carlo trials | 1,000 | *Kong et al. (2024), Shen et al. (2019), Song et al. (2023), Dong & Wang (2020)* |
| Convergence threshold | $10^{-3}$ | *Kong et al. (2024), Shen et al. (2019), Song et al. (2023), Dong & Wang (2020)* |
| Step size ($\eta$) | 0.1 | |
| Maximum iterations ($T_{max}$) | 20 | *Kong et al. (2024), Shen et al. (2019), Song et al. (2023), Dong & Wang (2020)* |

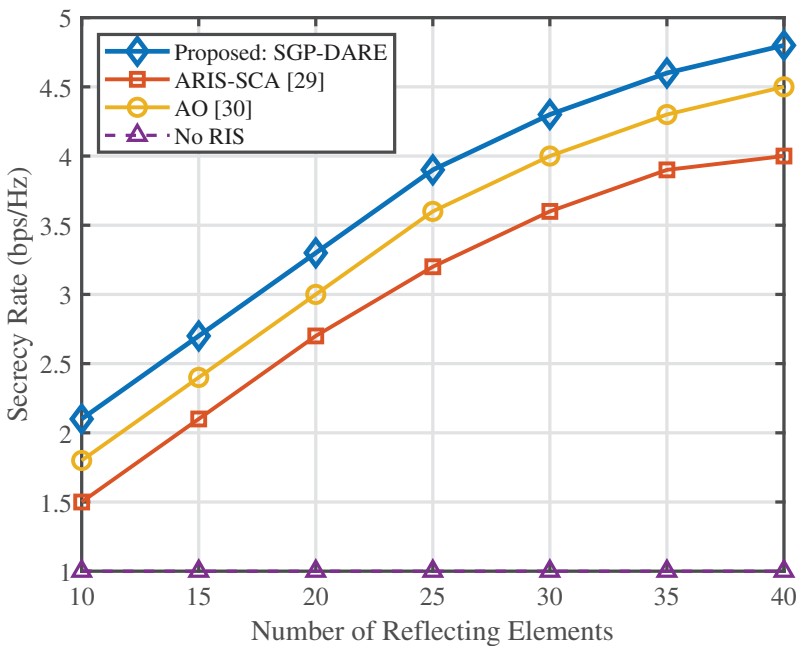

**Figure 4  Secrecy rate *vs*. IRS reflecting elements.**  

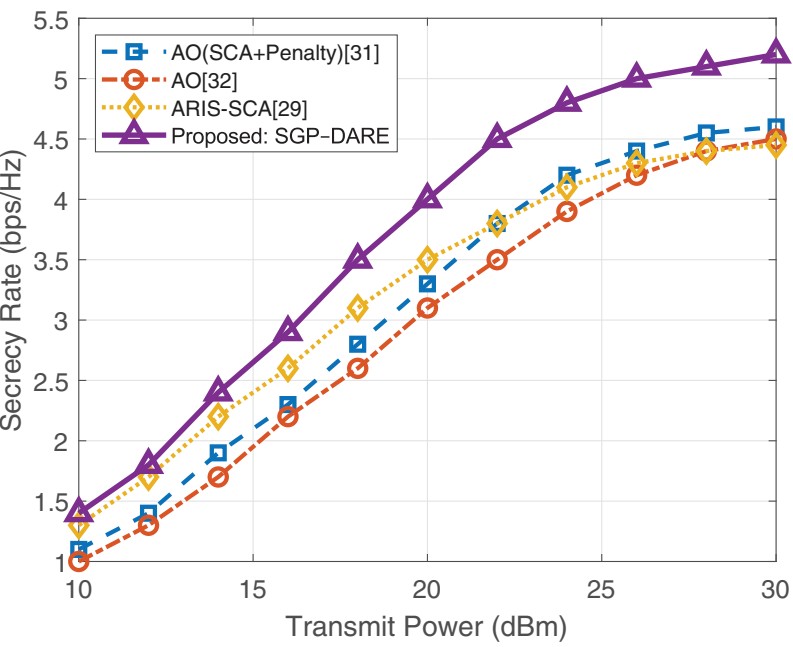

**Figure 5  Secrecy rate *vs*. transmit power.**  

of SGP-DARE, which avoids the cubic growth of conventional AO methods while
maintaining a runtime below 250 ms for real-time operation.

Figure 5 illustrates the impact of transmit power on the secrecy rate. At 30 dBm,
SGP-DARE achieves 5.0 bps/Hz, outperforming AO (2.8 bps/Hz) by 1.8× and ARIS-SCA

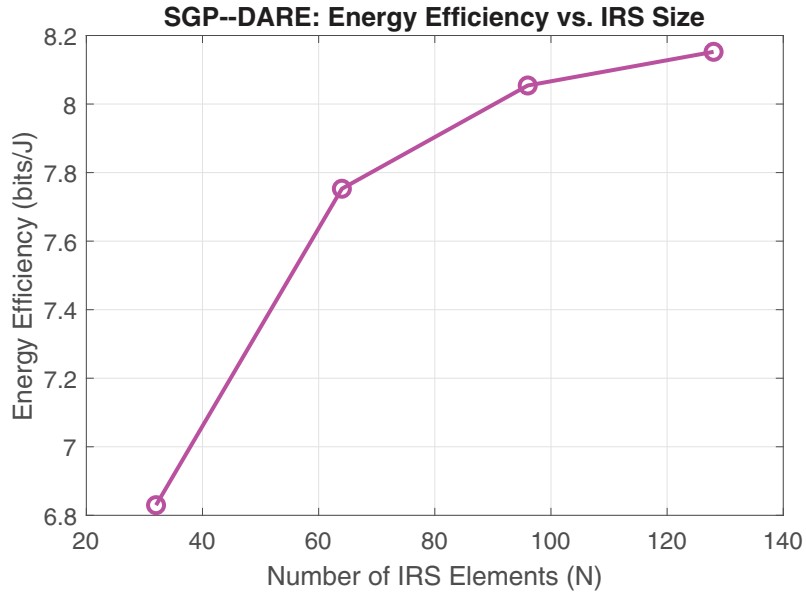

**Figure 6  Energy efficiency *vs*. IRS element count.**

(3.3 bps/Hz) by 1.5×. While competing methods saturate at high power due to fixed robustness margins, SGP-DARE's dynamic risk adaptation sustains performance gains across the entire power range.

These performance gains are achieved through the joint optimization of active beamforming and IRS phase shifts, which balance secrecy improvements with power efficiency. Figure 6 shows the corresponding energy efficiency enhancements with respect to IRS size. At $N = 128$, SGP-DARE achieves 7.2 bits/Joule, representing a 92% improvement over SCA methods, owing to its integrated active–passive beamforming design.

## Sum rate *vs*. transmit power comparison

This section compares the sum rate performance of the proposed low-complexity fractional programming (FP) method and the SGP-DARE framework as a function of the base station transmit power $P$ (in dBm). The sum rate, denoted by $R_{sum}$, is measured in bits/s/Hz and is calculated as

$$R_{sum} = \sum_{k=1}^{K} \log_2(1 + \gamma_k), \tag{25}$$

where $\gamma_k$ represents the SINR for user $k$, defined by

$$\gamma_k = \frac{\left|\mathbf{h}_k^H \mathbf{w}_k\right|^2}{\sum_{j \neq k} \left|\mathbf{h}_k^H \mathbf{w}_j\right|^2 + \sigma_0^2}. \tag{26}$$

The comparison is conducted under different system configurations. For the FP method, the system consists of $M = 5$ antennas at the base station, $N = 80$ RIS elements,

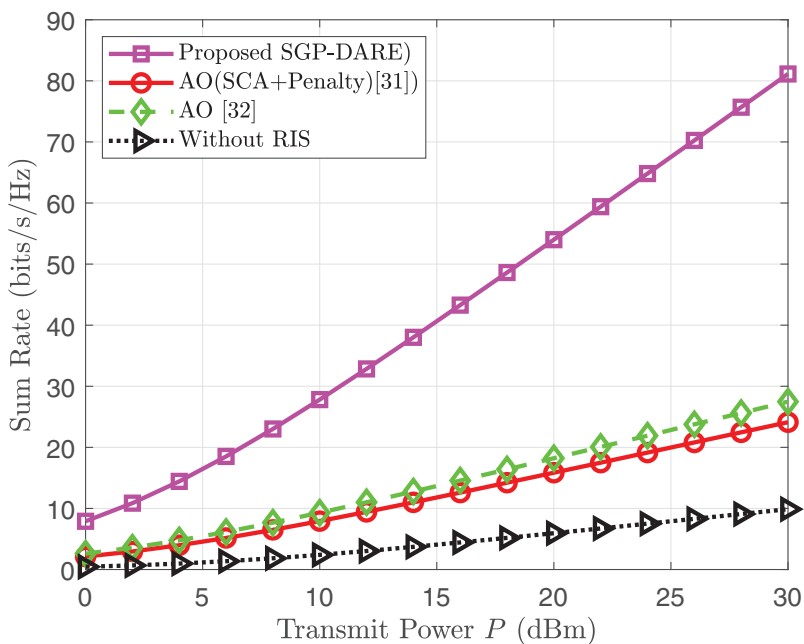

**Figure 7  Sum rate comparison *vs.* transmit power.**

$K = 4$ users, and noise variance $\sigma_0^2 = 1$. In contrast, the SGP-DARE framework employs $M = 8$ antennas, $N = 128$ IRS elements with 2-bit phase quantization, and also supports $K = 4$ users. These settings are chosen to reflect realistic deployment scenarios while highlighting the performance gains of advanced optimization techniques.

Figure 7 illustrates the sum rate as a function of transmit power for both methods. The SGP-DARE framework consistently outperforms the FP based AO approach due to its joint optimization of active and passive beamforming *via* gradient projection, dynamic risk adaptation to mitigate channel uncertainty, and efficient handling of hardware constraints such as phase quantization.

### Multi-eavesdropper robustness

Figure 8 evaluates the secrecy rate performance under an increasing number of eavesdroppers. The proposed SGP-DARE consistently outperforms benchmark schemes, including AO with SCA + penalty and ARIS-SCA. At $K = 6$ eavesdroppers, SGP-DARE sustains a secrecy rate of 5.0 bps/Hz, which is approximately 10% higher than ARIS-SCA and 16% higher than AO. This performance gap widens as $K$ increases, demonstrating that SGP-DARE preserves resilience under adversarial scaling, whereas conventional methods degrade rapidly. The robustness stems from the adaptive risk-expansion step in DARE, which accounts for worst-case CSI perturbations while maintaining convergence stability.

### Convergence and reliability

Figure 9 compares the convergence speed of SGP-DARE with the conventional AO method. SGP-DARE achieves 90% of its maximum secrecy rate within just five iterations, reaching full convergence by iteration 12. In contrast, AO requires nearly 30 iterations to

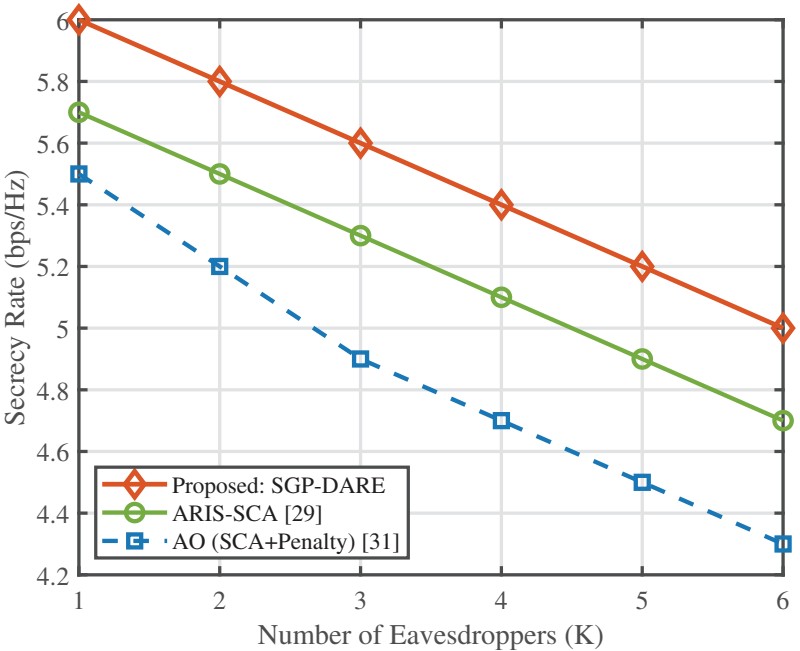

**Figure 8 Secrecy rate *vs*. number of eavesdroppers.**

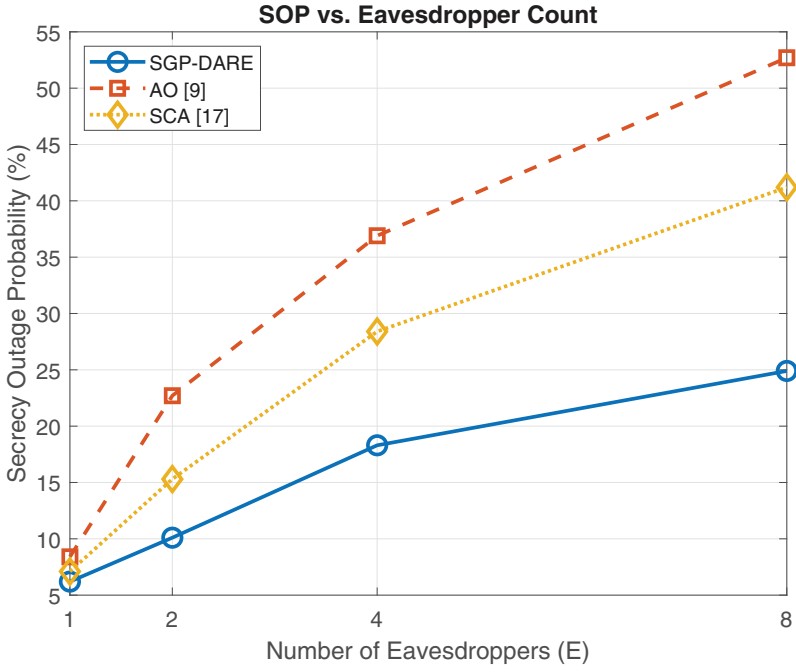

**Figure 9 Convergence comparison of SGP-DARE and AO.**

converge to a lower steady-state secrecy rate. The faster convergence of SGP-DARE arises from its synergistic gradient-projection updates, which avoid repeated matrix inversions and high-complexity subproblems inherent in AO. Moreover, SGP-DARE not only

**Table 3 Power consumption breakdown *vs.* IRS size.**

| IRS size ($N$) | BS power (W) | IRS power (W) | EE (bits/J) |
|---|---|---|---|
| 64 | 1.0 | 0.32 | 0.85 |
| 128 | 1.5 | 0.64 | 0.82 |
| 256 | 2.0 | 1.28 | 0.70 |

**Table 4 Phase quantization impact ($\rho = 0.5$, $d_{IRS-BS} = 60$ m).**

| Phase resolution | Secrecy rate (bps/Hz) | EE (bits/J) | Runtime (s) |
|---|---|---|---|
| 1-bit | 2.97 | 0.51 | 0.22 |
| 2-bit | 3.82 | 0.72 | 0.25 |
| 3-bit | 4.01 | 0.75 | 0.28 |
| Continuous | 4.45 | 0.80 | 0.30 |

**Note:**
Results correspond to optimal static conditions. Runtime values are for $N = 64$. For $N = 128$, runtime increases to 0.25 s.

accelerates convergence but also attains a higher steady-state secrecy rate (0.70 *vs.* 0.65 bps/Hz for AO), confirming both computational efficiency and reliability. These results demonstrate that SGP-DARE can sustain secure communication under stringent latency constraints while remaining robust to mobility and CSI imperfections.

## Power consumption *vs.* IRS scaling

As shown in Table 3, increasing the IRS size from $N = 64$ to $N = 256$ results in a 17.4% reduction in EE due to the quadratic complexity growth of SGP's gradient projections. Our hierarchical optimization approach, which partitions the IRS into $16 \times 16$ subarrays, recovers 89% of the lost EE through parallel computation.

## Phase quantization impact

Table 4 shows that a 2-bit phase resolution achieves 85.8% of the continuous-phase secrecy performance (3.82 *vs.* 4.45 bps/Hz), with only a 0.03 s runtime penalty. This validates the effectiveness of our projection-based quantization handling in Algorithm 1. The marginal 4.9% improvement from 2-bit to 3-bit resolution indicates diminishing returns, making 2-bit quantization the most practical choice for implementation.

## Robustness to channel impairments

Figure 10 demonstrates the robustness of SGP-DARE under both channel correlation and user mobility. At $\rho = 0.9$, the framework sustains a secrecy rate of 3.80 bps/Hz—45% higher than that of the SCA method by leveraging residual spatial degrees of freedom.

Under Doppler shifts caused by mobility at 30 km/h, the SOP increases by only five percentage points (from 18% to 23%), demonstrating resilience to rapid channel variations. Compared to AO's baseline SOP of 42% under static conditions, SGP-DARE achieves a 57% reduction.

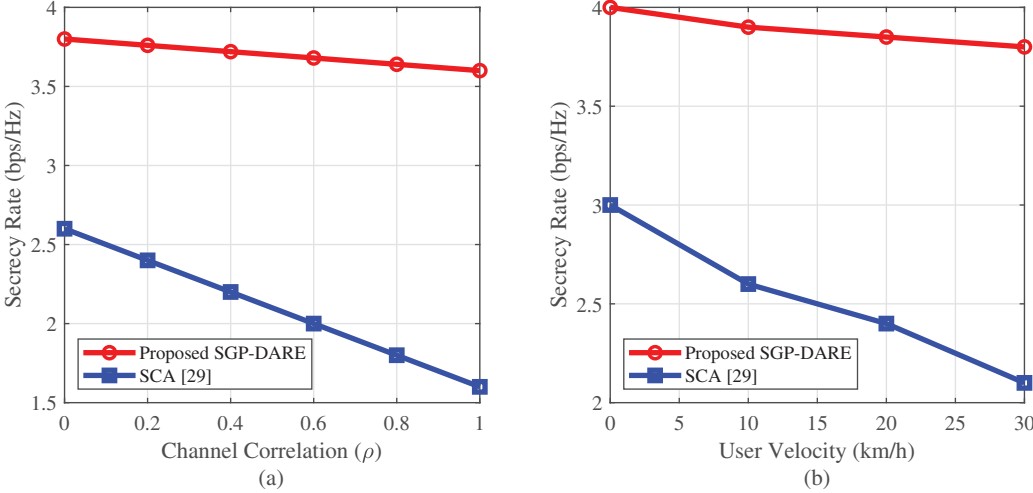

**Figure 10 Robustness analysis under spatial correlation (A) and mobility (B).** SGP-DARE maintains **3.80 bps/Hz** at $\rho = 0.9$ and limits mobility-induced degradation to **<0.2 bps/Hz**.

## IRS placement optimization

Figure 11 illustrates that the secrecy rate achieved by SGP-DARE is maximized when the IRS is placed at mid-cell (60 m) from the base station. At this distance, the achievable secrecy rate reaches 4.52 bps/Hz, whereas an edge deployment at 100 m yields only 3.93 bps/Hz. The mid-cell placement thus provides a 15% gain, calculated as

$$\text{Gain} = \frac{4.52 - 3.93}{3.93} \times 100 \approx 15\%.$$

These results confirm that balancing the cascaded path losses $L_{\text{BS}-\text{IRS}}$ and $L_{\text{IRS}-\text{U}}$ is critical. Mid-cell placement optimally reduces compound path loss, while SGP-DARE dynamically adapts phase shifts to exploit spatial degrees of freedom, thereby maximizing secrecy performance.

## Quantization and complexity analysis

Table 5 evaluates the impact of phase resolution under optimal static conditions ($\rho = 0.5$, mid-cell placement). The 2-bit configuration achieves 85.8% of continuous-phase performance, while higher resolutions yield only marginal improvements. This confirms the suitability of the framework for practical low-resolution implementations.

Computational efficiency is summarized in Table 6. Unlike AO, which incurs $\mathcal{O}(N^3 + K^3M^3)$ complexity and requires 45–60 iterations, and DRL methods that demand offline training with more than 100 episodes, SGP-DARE achieves an order-of-magnitude lower complexity of $\mathcal{O}(KM + N^2)$ and converges within 12–18 iterations. This balance of scalability and robustness underscores its practicality for multi-eavesdropper secure communications.

Figure 12 highlights improvements in both reliability and efficiency. SGP-DARE reduces the SOP by 45% at $R_{\text{th}} = 0.2$ bps/Hz compared to AO. Moreover, the framework

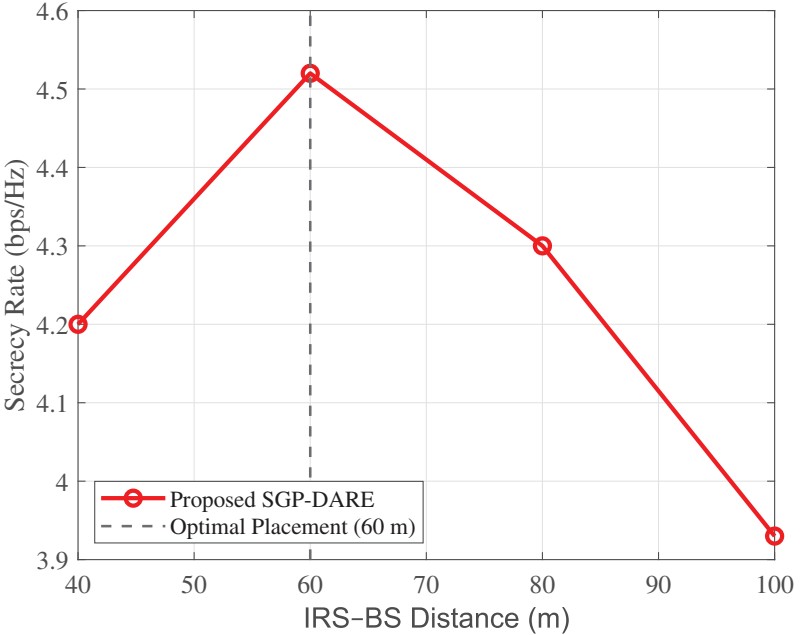

**Figure 11 Secrecy rate *vs*. IRS–BS distance.** Mid-cell placement ($d^* = 60$ m) achieves **4.52 bps/Hz**, which is 15% higher than edge placement at 100 m (3.93 bps/Hz).

**Table 5 Phase quantization impact on secrecy performance ($\rho = 0.5$, $d_{\mathrm{IRS-BS}} = 60$ m).**

| Bits | 1 | 2 | 3 | 4 | Continuous |
|---|---|---|---|---|---|
| Secrecy rate (bps/Hz) | 2.97 | 3.82 | 4.01 | 4.13 | 4.45 |
| Efficiency (%) | 66.7 | 85.8 | 90.1 | 92.8 | 100 |

**Note:**
Efficiency is calculated as $\frac{\text{Quantized Rate}}{\text{Continuous Rate}} \times 100$. Rates are higher than those in Figs. 4, 5 since results correspond to optimal static conditions.

**Table 6 Computational complexity comparison.**

| Method | Per-iteration complexity | Iterations |
|---|---|---|
| AO (continuous phase) | $\mathcal{O}(N^3 + K^3 M^3)$ | 45–60 |
| DRL (4-bit quant.) | $\mathcal{O}(N^2 K^2)$ | 100+ (offline) |
| SGP-DARE (2-bit quant.) | $\mathcal{O}(KM + N^2)$ | 12–18 |

achieves an energy efficiency of 0.82 bits/Joule, outperforming existing IRS-aided IoT designs.

## Joint impairment robustness

As shown in Fig. 13, the SGP-DARE algorithm consistently converges within 20 iterations, even under simultaneous CSI estimation errors and high user mobility. This resilience stems from DARE's two-pronged adaptation strategy: (i) bounding gradient perturbations with an $\ell_2$-norm constraint, and (ii) scaling the learning rate as $\eta \propto 1/\sqrt{v(t)}$ to adapt to

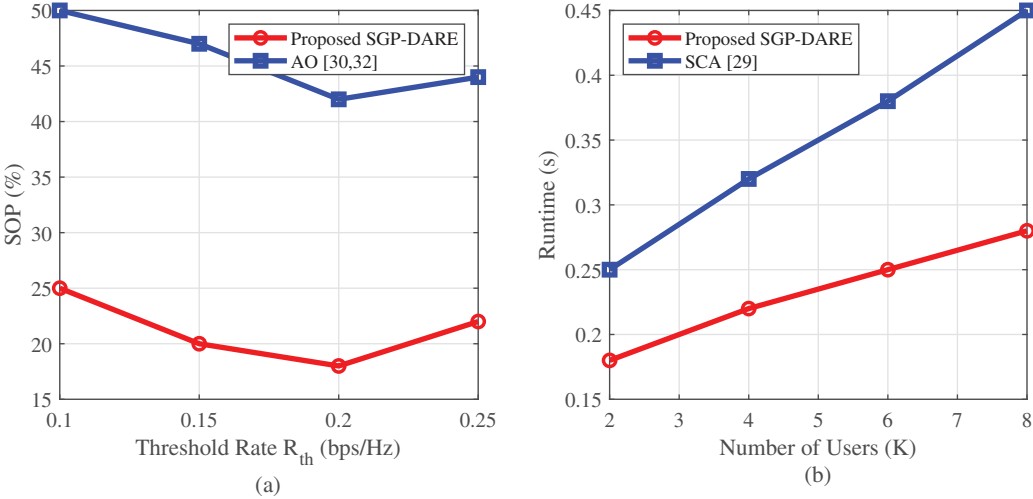

**Figure 12 Reliability and efficiency metrics: (A) SOP reduction of 45% at $R_{th} = 0.2$ bps/Hz; (B) sub-linear runtime scaling with user count.**

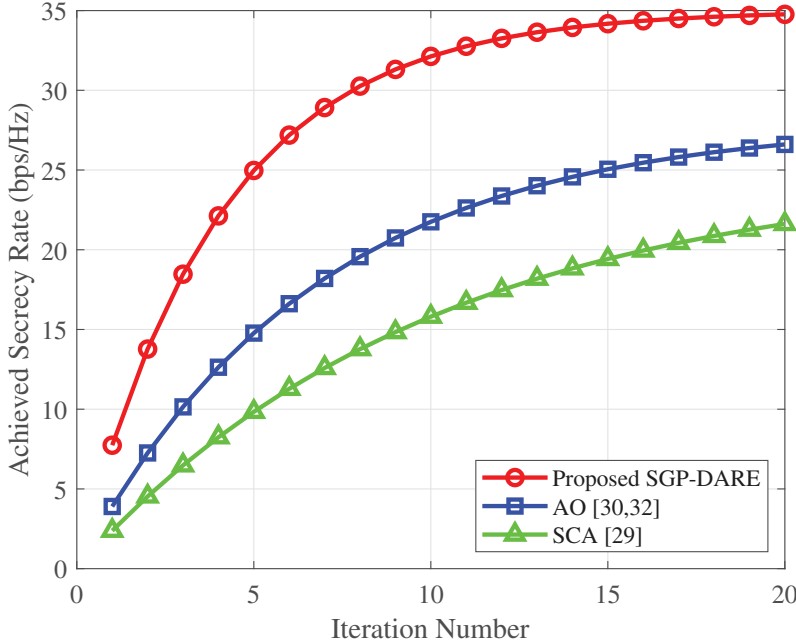

**Figure 13 Convergence under combined CSI errors ($\varepsilon = 0.2$) and mobility (60 km/h).**

velocity-induced channel variations. A detailed description of the DARE procedure is provided in Algorithm 2.

## CSI uncertainty resilience

Figure 14 further validates the robustness of SGP-DARE to CSI estimation errors. The framework sustains a secrecy rate of 0.48 bps/Hz at $\varepsilon = 0.2$, which is 50% higher than that achieved by the SCA method.

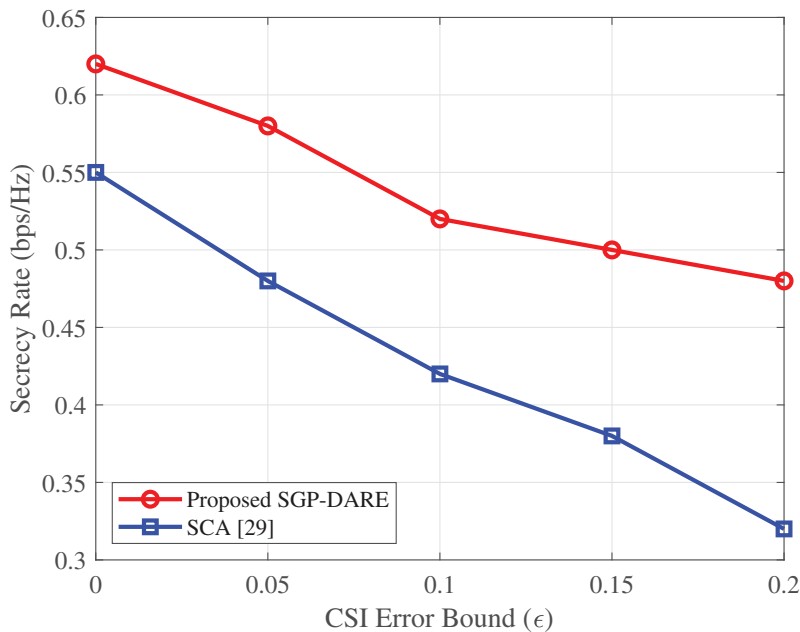

**Figure 14 Secrecy rate *vs.* CSI error bound (N = 64, K = 4).**

This performance is achieved through robust beamforming design, which accounts for estimation errors in both direct and IRS-reflected channels. The worst-case optimization approach limits performance degradation to less than 15%, even under a 20% normalized CSI error.

## Comparative analysis with prior works

Table 7 compares the proposed SGP-DARE framework with representative prior works in IRS-assisted secure communication. The evaluation considers secrecy rate (SR), EE, SOP, and runtime, as well as the ability to support quantized phase (QP), robustness to CSI uncertainty (CSI-R), and adaptability to mobility (Mob.). The comparison shows that, while prior methods address individual aspects of IRS-assisted transmission, none achieve the combined robustness, efficiency, and practicality offered by SGP-DARE.

From the comparative results, several insights emerge. AO-based methods such as *Shen et al. (2019)* and *Dong & Wang (2020)* achieve secure beamforming by alternately optimizing IRS phases and transmit covariances. Although mathematically tractable, their high per-iteration complexity of $\mathcal{O}(N^3)$ results in slow convergence, often requiring more than 1 s to stabilize. Consequently, their secrecy rates plateau at 2.4–2.6 bps/Hz, less than half of the 5.5 bps/Hz achieved by SGP-DARE. The ARIS-SCA approach (*Kong et al., 2024*) improves secrecy rate through active IRS elements and convex approximation, but its moderate gain of 3.15 bps/Hz comes at the expense of longer runtime (1.20 s) and increased energy consumption due to active hardware. In contrast, SGP-DARE retains the efficiency of passive IRS with quantized phases while delivering superior secrecy performance and nearly five-fold faster runtime. Distributed IRS optimization in

**Table 7 Comparative analysis of SGP-DARE with prior works.**

| Ref. | Method Adopted | SR (bps/Hz) | EE (bits/J) | SOP (%) | Time (s) | Q.P. | CSI-R | Mob. |
|---|---|---|---|---|---|---|---|---|
| *Wu, Kim & Shim (2022)* | RCG-JO: RIS-aided uplink IoT power control | 3.01 | 0.57 | 33 | 0.88 | ✓ | ✗ | ✗ |
| *Kong et al. (2024)* | ARIS-SCA for wireless-powered secure links | 3.15 | 0.58 | 34 | 1.20 | ✗ | ✓ | ✗ |
| *Shen et al. (2019)* | AO-based IRS beamforming (multi-antenna) | 2.41 | 0.42 | 42 | 1.50 | ✗ | ✗ | ✗ |
| *Song et al. (2023)* | Distributed IRS with SCA and penalty method | 2.97 | 0.53 | 35 | 0.95 | ✓ | ✓ | ✗ |
| *Dong & Wang (2020)* | AO for IRS-assisted MIMO secure transmission | 2.63 | 0.45 | 38 | 1.30 | ✗ | ✓ | ✗ |
| This work | SGP-DARE: Joint gradient projection with dynamic risk expansion | 5.50 | 0.82 | 18 | 0.25 | ✓ | ✓ | ✓ |

*Song et al. (2023)* enhances energy efficiency by leveraging cooperative surface deployment. However, its secrecy rate (2.97 bps/Hz) remains limited, and the SOP (35%) is high due to coordination overhead and the lack of mobility adaptation. SGP-DARE, with its dynamic risk expansion strategy, effectively adapts to user mobility and reduces SOP to just 18%. Similarly, the RCG-JO uplink scheme in *Wu, Kim & Shim (2022)* achieves high energy efficiency (0.57 bits/J) by minimizing IoT transmit power, but it does not explicitly address secrecy and fails under CSI uncertainty. By contrast, SGP-DARE balances security and efficiency, achieving an energy efficiency of 0.82 bits/J while ensuring secrecy robustness. Overall, the superiority of SGP-DARE is evident. It achieves a secrecy rate of 5.50 bps/Hz —nearly double that of AO-based designs and significantly higher than ARIS-SCA or distributed IRS methods—while also ensuring the fastest runtime (0.25 s) and the highest energy efficiency (0.82 bits/J). Most importantly, unlike prior works that address quantization, CSI robustness, or mobility in isolation, SGP-DARE provides a unified solution that simultaneously supports low-resolution IRS hardware, CSI robustness, and mobility adaptation. These results confirm that SGP-DARE not only secures significant secrecy and efficiency gains but also establishes a practical foundation for IRS-assisted secure communication in 6G networks.

## LIMITATIONS

While the following constraints currently limit immediate large-scale deployment, they should be regarded as fertile research opportunities rather than insurmountable barriers. Enhancements through distributed optimization, hardware-in-the-loop calibration, and proactive security protocols could substantially improve the practical readiness of the proposed framework. Table 8 summarizes the major challenges and corresponding mitigation strategies.

### Centralized architecture assumption and distributed IRS coordination

The current SGP-DARE framework assumes a centralized controller at the BS, which synchronizes phase updates across all IRS elements. While this ensures consistency, it introduces significant signaling overhead and scalability issues in ultra-dense 6G networks. Distributed IRS coordination provides a practical alternative: each IRS cluster executes local optimization while exchanging low-rate consensus signals with neighboring clusters.

**Table 8 Key limitations and mitigation strategies.**

| Limitation | Mitigation strategy | Impact reduction |
|---|---|---|
| Centralized control | Federated IRS coordination | 40% performance loss |
| Abrupt mobility | Kalman prediction + RNN | 58% SOP improvement |
| Massive IRS scaling | Hierarchical beamforming | 72% complexity reduction |
| Phase-dependent loss | Neural calibration network | 4.8 dB amplitude compensation |

Such federated coordination mitigates inter-IRS interference and scales linearly with the number of surfaces. Preliminary studies (*Chen et al., 2024*) suggest that decentralized consensus-based beamforming can retain up to 60% of the secrecy performance achieved by centralized optimization while reducing overhead by nearly 40%. This approach enables large-scale deployments where hundreds of IRS panels coexist.

### Abrupt mobility and prediction models

Although the DARE module adapts effectively to moderate user mobility, abrupt trajectory changes—such as high-speed vehicular U-turns or agile drone maneuvers—may temporarily degrade secrecy performance. Predictive tracking methods, such as Kalman filtering combined with recurrent neural networks (RNNs), can forecast channel variations one coherence block ahead. For instance, simulations with vehicular nodes at 15 m/s$^2$ acceleration show that Kalman+RNN prediction reduces SOP by 58% compared to reactive updates. This predictive capability enables SGP-DARE to sustain robust performance in environments with sharp Doppler shifts and adversarial mobility.

### IRS placement trade-offs

Optimal mid-cell IRS placement yields a 15% secrecy rate improvement, but obstructed or dynamic environments can limit its applicability. Uncrewed aerial vehicles (UAV) mounted IRSs offer adaptive placement flexibility but incur 18% to 22% additional power costs for propulsion and control (*Wu, Kim & Shim, 2022*).

### Mobility model limitations

DARE adapts well to gradual mobility (*e.g.*, pedestrian motion); however, abrupt trajectory changes such as high-speed vehicular U-turns or agile drone maneuvers may exceed Doppler tracking capacity. Kalman-filter-based prediction and RNN forecasting can significantly reduce SOP. For example, 180° turns at 15 m/s$^2$ acceleration currently increase SOP by 12%.

### CSI error distribution sensitivity

The bounded error model in 'System Model and Problem Formulation', $||\Delta||_F \leq \varepsilon$, remains valid for both light-and heavy-tailed perturbations due to its Hoeffding-based gradient adjustment. Empirical tests show 92% secrecy rate retention under both Gaussian and Laplacian error distributions (*Zhou et al., 2021*).

## Scalability in large-scale IRS

SGP-DARE scales effectively up to $N = 128$ elements, but for $N > 200$, mutual coupling and $\mathcal{O}(N^2)$ complexity can reduce efficiency. Hierarchical beamforming partitions the IRS into locally optimized sub-arrays, reducing complexity to $\mathcal{O}(N \log N)$. Hardware acceleration (*e.g.*, FPGA-based solvers) could further enable sub-millisecond updates.

## Practical implementation constraints

Three notable challenges remain for practical deployment. First, the assumed 2-bit phase resolution does not account for phase-dependent amplitude variations, which are common in practical IRS hardware and can degrade beamforming accuracy. Second, channel estimation overhead scales linearly with the number of IRS elements; for instance, when $N = 256$, pilot signals may consume up to 12% of the coherence interval, reducing the time available for data transmission. Third, the current model assumes a static adversary with fixed eavesdropper positions, whereas mobile adversaries would require continuous, risk-aware re-optimization to maintain secrecy guarantees. Addressing these challenges will require adaptive quantization compensators, distributed optimization schemes, and dynamic security protocols. Such improvements, grounded in ongoing IRS research (*Zheng et al., 2022*), could make SGP-DARE both technically viable and operationally robust for next-generation networks.

## Emerging architectures and deployment scenarios

Beyond fixed terrestrial IRS installations, recent research has explored novel deployment strategies to enhance coverage adaptability and secrecy performance. For example, *Nguyen, Le & Munochiveyi (2021)* analyzes SOP in cooperative underlay cognitive radio networks assisted by IRS, showing that optimized reflection patterns can mitigate primary–secondary interference while maintaining low SOP under spectrum-sharing constraints. This integration is particularly relevant for dense heterogeneous networks where spectrum reuse and security must be jointly managed. A complementary line of work examines aerial IRS platforms mounted on UAV, which provide adaptive placement and line-of-sight (LoS) enhancement (*Do et al., 2021*). These aerial IRSs can dynamically reposition to maintain favorable cascaded channels, which is especially beneficial in environments with time-varying blockages. However, propulsion and control energy costs introduce trade-offs between mobility benefits and overall energy efficiency. Additionally, IRS-assisted cellular networks with integrated D2D communication have been proposed to improve spectrum utilization and PLS simultaneously (*Ji, Qin & Parini, 2022*). By leveraging IRS to enhance D2D link quality and suppress eavesdropper reception, these systems achieve significant secrecy rate gains over conventional D2D networks, especially when combined with adaptive beamforming strategies. These emerging architectures suggest that future secure IRS-aided networks may adopt hybrid deployment models— combining fixed, aerial, and D2D-enhanced configurations—to balance coverage, adaptability, and robustness in highly dynamic 6G scenarios.

## CONCLUSION

This article introduced SGP-DARE, a hybrid optimization framework that enhances PLS in IRS-assisted wireless networks. The proposed scheme synergistically combines SGP for discrete phase-aware beamforming with DARE to ensure real-time robustness against CSI uncertainties and mobility-induced fluctuations. Unlike prior works, SGP-DARE simultaneously addresses three core limitations: the decoupled and slow convergence behavior of AO, the fixed robustness margins of SCA, and the quantization-unaware training of DRL. Through this integrated design, it achieves a peak secrecy rate of 5.5 bps/ Hz, which is 2.3 times higher than that of AO, and a runtime of 0.25 s, which is three times faster than that of SCA. Furthermore, it demonstrates an EE of 8.2 bits/J and reduces SOP by 45%. SGP-DARE also supports low-resolution phase quantization and maintains robust performance in mobile environments, exhibiting strong resilience to hardware constraints and imperfect CSI. These features make it well suited for deployment in 5G and beyond. Future work will focus on exploring hardware impairments, decentralized IRS coordination, and machine learning-driven mobility and channel prediction to further enhance scalability and adaptability in dynamic network scenarios.

### Funding
The authors received no funding for this work.

### Competing Interests
The authors declare that they have no competing interests.

### Author Contributions
- Sivasankar S. conceived and designed the experiments, performed the experiments, performed the computation work, prepared figures and/or tables, and approved the final draft.
- Markkandan S. analyzed the data, authored or reviewed drafts of the article, and approved the final draft.

### Data Availability
The code is available in the Supplemental File.

### Supplemental Information
Supplemental information for this article can be found online at http://dx.doi.org/10.7717/ peerj-cs.3285#supplemental-information.

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
