# Peer review of "A hybrid algorithmic model for enhancing security in intelligent reflecting surface-assisted wireless communication"

_PeerJ Computer Science, doi:10.7717/peerj-cs.3285_

## Round 0.1 · original submission · Major Revisions

· Academic Editor

Major Revisions

See the comments from two reviewers and please revise your paper accordingly.

**Language Note:** The review process has identified that the English language must be improved. PeerJ can provide language editing services - please contact us at [email protected] for pricing (be sure to provide your manuscript number and title). Alternatively, you should make your own arrangements to improve the language quality and provide details in your response letter. – PeerJ Staff

Reviewer 1 ·

Basic reporting

This paper introduced the Hybrid Gradient Descent Optimization (HGDO) framework to address the critical challenges of secure communication in IRS-aided multi-user systems. Although the studied the topic has been extensively studied recently. The paper proposed a new optimization method that has been shown to be effective.

However, the reviewer has the following concerns:
1) As mentioned in the introduction, the topic has been extensively discussed in the existing papers. However, the authors need to provide extensive simulation results to show the advantages of the proposed methods.

2) The assumption of the availability of CSI is too strict. The authors need to mention the related channel estimation for RIS-aided systems, such as Channel estimation for RIS-aided multiuser millimeter-wave systems

3) The authors need to analyze the computational complexity of the proposed method.

4) Although the authors mentioned the robust design, they did not mention the technical methods to deal with the imperfect CSI.

5) The introduction is not complete, and lacks some important tutorial papers about RIS to help readers understand the recent advances in the area of RIS.

Experimental design

-

Validity of the findings

-

Reviewer 2 ·

Basic reporting

-

Experimental design

-

Validity of the findings

-

Additional comments

This paper studies adaptive Hybrid Optimization for IRS-Assisted Secure Wireless Networks. The reviewer has the following concerns:

1) The conclusions drawn in the abstract are not new, as many works suggest. Please explain in detail the contribution of this paper and the difference from existing work.

2) The contribution is not clear as well. Please reconsider the layout.

3) The operation of the IRS requires energy. In the considered system, how can operating energy be provided for multiple IRSs?

4) There is a lot of research on these topics. I think the authors did not provide a detailed survey of existing works, which casts doubt on the novelty and contributions of this work.

5) There are many typos; please carefully check throughout the paper.

---

## Round 0.2 · Major Revisions

· Academic Editor

Major Revisions

The previous reviewers were unable to re-review so I had to seek a new reviewer. Please revise your paper according to the comments from R3.

**Language Note:** The review process has identified that the English language must be improved. PeerJ can provide language editing services - please contact us at [email protected] for pricing (be sure to provide your manuscript number and title). Alternatively, you should make your own arrangements to improve the language quality and provide details in your response letter. – PeerJ Staff

Reviewer 3 ·

Basic reporting

In the Abstract, you should not put numerous values, but only basic principles and gains. Numerical data should be given in the section with analysis of the results.

Use the abbreviations once introduced and do not define them again (such as CSI, IRS, SGP, DARE, ... in the following sections and in the Conclusion). IRS is defined again in section 3, etc.
On the other hand, DFT, DQN, etc. are not defined at all, nor are defined the abbreviations used in the tables!

The description of the contents of the sections does not correspond to the actual situation.
Subsections 2.4 and 2.7 and Tables 1 and 2 therein are premature. The comparison should be made much later, when analyzing the results.
2.6 should be in the introduction, not in section 2 (Related Works)

The sentence:
"User mobility is modeled with velocities v ≤ 30 km/h, estimated via Doppler shift tracking"
is repeated (lines 175 and 185), and the claim is supported by two different references. Which one is appropriate?

The authors must provide references for any formulas they did not derive by themselves in this paper!
At the end of the formula, put a period or a comma, depending on the position in the sentence.
All variables in equations must be defined, and different quantities cannot be denoted by the same letter (like eta)

2.3× improvement ... and 1.7× gain over =>
2.3 times improvement ... and 1.7 times gain over
etc.

User mobility causes fading, which should also be taken into consideration.

Capitalize CSI and other abbreviations in references!

References are fresh, but they should be listed in the order in which they appear in the text!

Experimental design

no comment'

Validity of the findings

no comment

---

## Round 0.3 · accepted · Accept

· Academic Editor

Accept

Accept. Please contact copy editors for further issues.

Reviewer 3 ·

Basic reporting

It is evident that the authors made an effort to respond to the remarks of the reviewers, and the paper can be accepted for publication after possibly necessary minor corrections.

Experimental design

no comment

Validity of the findings

no comment